Sunset Yellow and Allura Red modulate Bcl2 and COX2 expression levels and confer oxidative stress-mediated renal and hepatic toxicity in male rats

Khayyat Latifa I. 1
Essawy Amina E. 2
Sorour Jehan M. 2
Soffar Ahmed ahmedsoffar@gmail.com ahmedsoffar@alexu.edu.eg 3
1 Biology Department, Faculty of Applied Sciences, Umm Al-Qura University , Mecca , Saudi Arabia
2 Zoology Department, Faculty of Science, Alexandria University , Alexandria , Egypt
3 Division of Molecular Biology, Zoology Department, Faculty of Science, Alexandria University , Alexandria , Egypt
Aziz Ramy
Electronic publication date: 2018 Sep 28
Publication date: 2018
Volume: 6
Electronic Location ID: e5689
Received 2018 Mar 19; Accepted 2018 Sep 3
Copyright: ©2018 Khayyat et al.
Copyright year: 2018
Copyright holder: Khayyat et al.
License: This is an open access article distributed under the terms of the Creative Commons Attribution License, which permits unrestricted use, distribution, reproduction and adaptation in any medium and for any purpose provided that it is properly attributed. For attribution, the original author(s), title, publication source (PeerJ) and either DOI or URL of the article must be cited.
License URL: https://creativecommons.org/licenses/by/4.0/

Keywords: Genotoxicity, Histopathology, Serum biochemistry, Sunset Yellow, Allura Red, COX2, Bcl2

Funding: Institute of Scientific Research and Revival of Islamic Heritage 43405033 This work was supported by the Institute of Scientific Research and Revival of Islamic Heritage at Umm-Al Qura University, project number 43405033. The funders had no role in study design, data collection and analysis, decision to publish, or preparation of the manuscript.

==============================
Studies on the adverse health effects caused by azo dyes are insufficient and quite contradictory. This work aims to investigate the possible toxic effect of two types of widely used food additives, Sunset Yellow and Allura Red, by assessing the physiological, histopathological and ultrastructural changes in the liver and kidney. Also, we investigated the genotoxic effect of both dyes on white blood cells. Thirty adult male albino rats were divided into three groups of 10 animals each: control (received water), Sunset Yellow-treated (2.5 mg/kg body weight) and Allura Red-treated (seven mg/kg body weight). The doses were orally applied for 4 weeks. Our results indicated an increase in the biochemical markers of hepatic and renal function (Aspartate aminotransferase, alanine aminotransferase, urea, uric acid and creatinine) in animals administered with the azo dyes. We also observed a noticeable increase in MDA and a marked decrease in total antioxidant levels in azo dye-treated animals compared to controls. Conversely, both dyes adversely affected the liver and kidney of albino rats and altered their histological and fine structure, with downregulation of Bcl2 and upregulation of COX2 expression. Our comet assay results showed a significant elevation in the fold change of tail moment in response to application of Sunset Yellow but not Allura Red. Collectively, we show that Sunset Yellow and Allura Red cause histopathological and physiological aberrations in the liver and kidney of male Wistar albino rats. Moreover, Sunset Yellow but not Allura Red induces a potential genotoxic effect.

Introduction

Synthetic food dyes are currently considered one of the most dangerous food additives. Their use in food stuffs raises a series of doubts regarding their cytotoxicity, due to the limited work evaluating the cytotoxicity of these compounds (Feng, Cerniglia & Chen, 2012). Nonetheless, the possible harmful effect of synthetic food colouring additives is a subject of public health concerns and has a critical attitude toward their use in food  preparation.

In general, synthetic azo dyes are considered the most versatile class of dyes, accounting for about 50% of the dyes produced every year (Puvaneswari, Muthukrishnan & Gunasekaran, 2006). The toxic adverse effects of the azo dyes are not a result of the native dye but the toxic derivatives, such as aniline, toluene, benzidine and naphthalene, formed during the dye degradation process (Gomes et al., 2013). The impacts of degraded products of azo dyes on human health have caused concern over the last few years. Some azo dyes have been linked to an increased incidence of bladder cancer, splenic sarcomas and hepatocarcinomas in humans and animals (Puvaneswari, Muthukrishnan & Gunasekaran, 2006).

Sunset Yellow and Allura Red are among the most commonly used azo dyes in the food industry worldwide. These dyes are used in foodstuffs, pharmaceutical products and cosmetics, as yellow and red colouring agents (Hashem et al., 2010). The current literature on the adverse health effects caused by Sunset Yellow and Allura Red is insufficient and quite contradictory. For example, Sasaki et al. (2002) reported that the application of Sunset Yellow did not induce cellular aberrations in stomach cells of mice that orally received the maximum allowed dose. Conversely, other research suggested that administration of Sunset Yellow causes mutagenic or carcinogenic effects in human (Sarhan, Shati & Elsaid, 2014). Histopathological alterations in the testes and various brain regions of rats have also been demonstrated, as a consequence of dietary exposure to Sunset Yellow (Mathur et al., 2005; Khiralla, Salem & El-Malky, 2015). Interestingly, Allura Red has been previously classified in one study as a non-genotoxic compound (Combes & Haveland-Smith, 1982) while in another recent work, it has been evaluated as a genotoxic agent (Jabeen et al., 2013).

Due to these contradicting observations, additional investigations are necessary to assess the potential adverse effects of Sunset Yellow and Allura Red. Hence, in this work, we raised the question of whether applying low doses of Sunset Yellow and Allura Red would adversely affect the liver and kidney structure and function. We also evaluated the possible genotoxic potential of both dyes on white blood cells, using the single cell electrophoresis assay (comet assay).

Materials and Methods

Thirty Wistar albino adult male rats of about 150 g body weight (b wt) were used in this work. Rats were kept in our animal facility for about 1 week before starting the experiment, for acclimatisation to the laboratory conditions.

Animals were housed in small plastic cages, maintained under constant conditions and were fed on a standard basal diet, as previously described in Khayyat et al. (2017). Our experimental procedures were approved by the Menoufia University IACUC Committee for Care of Laboratory Animals (Approval No.: MNSP155).

Sunset Yellow (CAS 2783-94-0; 90% purity) and Allura Red (CAS 25956-17-6; 80% purity) were purchased from Sigma–Aldrich (Munich, Germany).

Experimental design

The animals were randomly divided into three groups (a control group and two treated groups) of 10 rats each, and were treated daily for 30 days, as follows:

Group 1: Animals were orally given distilled water (one ml/kg b wt.) and served as controls.

Group 2: Animals were orally given Sunset Yellow (2.5 mg/kg b wt.) dissolved in 1 ml of distilled water (Doguc et al., 2013).

Group 3: Animals were orally given Allura Red (seven mg/kg b wt.) dissolved in 1 ml of distilled water (Doguc et al., 2013).

At the end of the experiment, the animals from all three groups were sacrificed under anaesthesia. Blood samples were collected in sterile centrifuge tubes and allowed to clot. After centrifugation (3,500 rpm, 15 min), sera of different samples were separated, transferred into empty tubes and biochemically analysed.

For the comet assay, other blood samples were collected, to isolate leucocytes. Regarding the histological and immunohistochemical examinations, pieces of liver and kidney were fixed in Bouin’s fluid. For ultrastructure studies, small pieces of the same organs were immersed in 4F1G fixative.

Biochemical analysis

The serum levels of aspartate aminotransferase (AST), alanine aminotransferase (ALT), urea, uric acid, creatinine, nitric oxide (NO), total antioxidant capacity (TAC) and malondialdehyde (MDA) were estimated, as detailed by Khayyat et al. (2017).

Histological and ultrastructural studies

Fixed tissues were dehydrated, cleared and embedded in paraffin wax. Paraffin sections of five µm thick were collected onto glass slides, deparaffinised and stained with haematoxylin and eosin (Bancroft & Gamble, 2002).

For the ultrastructural investigation, tissues were prepared as indicated by Khayyat et al. (2017) before visualisation under a Jeol 100CX electron microscope.

Immunohistochemical study

Sections from the kidney were loaded on charged slides and incubated overnight at 37 °C, for immunolabelling. Primary antibodies of Bcl2 (Clone 100/D5, Thermo Scientific, Waltham, MA, USA) and COX2 (Thermo Scientific, USA) were applied for immuno-localisation using UltraVision ONE detection system (Thermo Scientific, USA), according to the manufacturer’s protocol.

Genotoxicity study

For the comet assay, leucocytes were isolated as specified by Khayyat et al. (2017). Briefly, blood samples were centrifuged for 5 min. The buffy coat layer of leucocytes was collected and frozen as aliquots of 250 µl in a cold freezing mixture (RPMI 1,640, 10% DMSO). Prepared samples were stored frozen at −80 °C until the comet assay was performed. About 20 ± 5 comets were scored per sample, and three different samples for each group were analysed. The tail moment was determined by the OpenComet software (Gyori et al., 2014).

Statistical analysis

The biochemical and comet assay data were expressed as mean ± SD of three to five replicates and were statistically evaluated by the Student’s t-test. The difference between samples was considered statistically significant at p < 0.05.

Results

Biochemical analysis

The administration of Sunset Yellow or Allura Red was associated with alterations in the levels of several biochemical parameters in the plasma of treated rats, as shown in Fig. 1. The plasma liver enzymes, AST and ALT, were significantly elevated (p < 0.05) in animals treated with either Sunset Yellow or Allura Red as compared to controls. The plasma concentrations of creatinine and urea were also significantly increased in response to the administration of the individual dyes, and both dyes induced a slight non-significant elevation in the level of uric acid compared to the control value. In addition, we observed a significant depletion (p < 0.05) in serum total antioxidants, accompanied by a significant elevation in plasma MDA and NO levels in all azo dye-treated animals as compared to the control.

Figure 1 Effect of Sunset Yellow and Allura Red on serum biochemistry and oxidative biomarkers in male rats.

(A) Bar graph showing the fold change in the level of measured parameters in response to administration of Sunset Yellow. (B) Bar graph showing the fold change in the level of measured parameters in response to administration of Allura Red. Control values are adopted for meta-analysis from Khayyat et al. (2017) and are normalized to 1 (Red line) for all measured parameters.

Histological and ultrastructural results

Regarding the liver sections of animals treated with the separate dyes, our results show that relative to the control, both dyes adversely affected the liver tissue of treated animals. Among the destructive effects in the liver, were the disorganisation of hepatic strands, as well as necrotic and hydropic degeneration of hepatic cells (Figs. 2A and 3A). Many hepatocytes were filled with vacuoles of variable size while others appeared with irregular-shaped or pyknotic nuclei (Figs. 2B and 3B). We also observed marked damage in the central vein region and congestion of blood sinusoids (Figs. 2A and 3A). Furthermore, we noticed remarkable leucocytes infiltration and an increased number of Kupffer cells in the liver sections of treated animals when compared to the controls.

Figure 2 Sunset Yellow induces histological and ultrastructural alterations in liver tissue of rat.

(A) LM of liver section from rats treated with Sunset Yellow showing disorganized hepatic strands, necrosis of most hepatocytes (H), congestion of blood vessel (star). H&E, Scale bar: 50 µm. (B) LM of liver section from rats treated with Sunset Yellow showing irregular shaped nuclei (N), thickened blood sinusoid with erythrocytes (star), leucocytic infiltration (I), vacuoles (V) and Kupffer cell (K). H&E, Scale bar: 20 µm. (C) EM of liver rats treated with Sunset Yellow showing cytoplasmic degeneration, electron dense mitochondria (M), clustered rER and pyknotic nuclei (N). Scale bar: 500 nm. (D) EM of liver rats treated with Sunset Yellow showing numerous glycogen particles (g), myelin bodies (my), collagen fibers (star). M, mitochondria; N, nucleus; L, lipid droplets. Scale bar: 500 nm.

Figure 3 Adverse effect of Allura Red on the histology and ultrastructure of liver tissue of rat.

(A) LM of liver section from rats treated with Allura Red showing hydropic degeneration in hepatocytes (H), damage in central vein (star), leucocytic infiltration (I). H&E, Scale bar: 50 µm. (B) LM of liver section from rats treated with Allura Red showing vacuoles in hepatocytes (V), irregular nuclei (N), Kupffer cell (K), leucocytic infiltration (I). H&E, Scale bar: 20 µm. (C) EM of liver rats treated with Allura Red showing disrupted hepatocyte, degenerated mitochondria (M), packed rER, lipid droplets (L) and irregular nucleus (N). Scale bar: 500 nm. (D) EM of liver rats treated with Allura Red showing altered mitochondria (M), disintegration rER, lipid droplets (L), pyknotic nucleus (N) and Kupffer cell (K). Scale bar: 500 nm.

Our electron micrographs revealed alterations in the regular structure of the hepatocytes of animals treated with Sunset Yellow or Allura Red. The cytoplasmic organelles were clumped and separated by rarified cytoplasm. Degenerated mitochondria and densely packed rough endoplasmic reticulum were also evident (Figs. 2C and 3C). Lipid droplets, fibrous tissue proliferation and myelin bodies were recorded (Figs. 2D and 3D). In addition, some nuclei of hepatocytes exhibited chromatin clumping, and some of these nuclei appeared pyknotic (Figs. 2C and 3D).

Histological examination of kidney tissue from rats administered with the individual dyes revealed several degenerative structural changes in the renal tubules and glomeruli. Vacuolisation, necrosis and sloughing of tubular epithelium, as well as infiltration of inflammatory cells in-between kidney tubules, were apparent (Figs. 4A and 5A). Some glomeruli were obliterated and had thickened Bowman’s capsules. Some Malpighian corpuscles appeared with lobulation and dilation of Bowman’s space, and vacuolation in the glomeruli and periglomerular haemorrhage were noted (Figs. 4B and 5B).

Figure 4 Sunset Yellow alters the normal architecture of kidney tissue of rat.

(A) LM of kidney section from rats treated with Sunset Yellow showing degenerated renal tubules (RT) and destructed glomeruli (G). H&E, Scale bar: 100 µm. (B) LM of kidney section from rats treated with Sunset Yellow showing vacuolar degenerated renal tubules (RT), vacuoles in glomerulus (G), leukocytes infiltration (I), large vacuole between the tubules (star), haemorrhage (h). H&E, Scale bar: 50 µm. (C) EM of kidney rat treated with Sunset Yellow showing proximal tubular cell with altered nucleus (N), condensed mitochondria (M), lysosomes (Ly), vacuoles (star), microvilli (mv). Scale bar: 500 nm. (D) EM of kidney rat treated with Sunset Yellow showing disrupted distal tubular cells with pyknotic nucleus (N), disordered mitochondria (M), vacuolated cytoplasm (star). Scale bar: 500 nm.

Figure 5 Allura Red induces histological and ultrastructural changes in kidney tissue of rat.

(A) LM of kidney section from rats treated with Allura Red showing necrosis of renal tubules (RT) and degenerated glomeruli (G). H&E, Scale bar: 100 µm. (B) LM of kidney section from rats treated with Allura Red showing vacuolated renal tubules (RT), dilated Bowman’s space (star), leukocytes infiltration (I), haemorrhage (h). H&E, Scale bar: 50 µm. (C) EM of kidney rat treated with Allura Red showing disrupted proximal tubular cells, irregular nucleus (N), disordered mitochondria (M), lysosomes (Ly), vacuoles (V), microvilli (mv). Scale bar: 500 nm. (D) EM of kidney rat treated with Allura Red showing disrupted distal tubular cells. N, nucleus; M, mitochondria; V, vacuoles. Scale bar: 500 nm.

Concerning the ultrastructure investigation of renal tubular cells from the kidney of animals treated with Sunset Yellow or Allura Red, most proximal and distal tubular cells revealed destroyed cytoplasm. Pleomorphic cytoplasmic vacuolation and loss of normal parallel basal arrangement of mitochondria in cells of proximal convoluted tubules were noticed (Figs. 4C and 5C). Most mitochondria were markedly condensed and had lost their cristae. Irregular-shaped dense lysosomes and pyknotic nuclei with condensed chromatin were seen in some of the proximal tubules (Figs. 4C and 5C). Meanwhile, many cells lining the distal tubules exhibited destroyed apical parts, electron-dense mitochondria, copious hydropic vacuoles and numerous lysosomes. The nuclei appeared irregular, small, atrophic or pyknotic (Figs. 4D and 5D).

Immunohistochemical analysis of Bcl2 and COX2 expression levels

Bcl2 expression levels were visibly reduced in kidney tissues of both Sunset Yellow- and Allura Red-treated animals in comparison to the controls (Figs. 6A–6C). Regarding COX2 expression, despite finding a discernible basal level in the controls, COX2 levels were remarkably increased in the kidney tissues of Allura Red-treated animals (Figs. 6D & 6F). The application of Sunset Yellow did not alter the COX2 expression levels relative to the control (Fig. 6E).

Figure 6 Immunohistochemical microscopic images showing the expression of Bcl2 and COX2 in the kidney of rat.

(A–C) Bcl2 expression levels in control, Sunset Yellow-, and Allura Red-treated animals. Bcl2 expressions are decreased in Sunset Yellow and Allura Red as compared to control. (A–F) COX2 expression levels in control, Sunset Yellow-, and Allura Red-treated animals. Application of Sunset Yellow did not alter COX2 levels as compared to control. However, application of Allura Red elevated the expression levels of COX2 specially in the uriniferous tubules and Malpighian corpuscles. Scale bar: 20 µm.

Genotoxicity

We investigated the integrity of the genomic content of leucocytes in response to the respective dyes, by using the comet assay. Our results showed a significant genotoxic effect induced in response to Sunset Yellow administration over the controls, as evidenced by an elevated tail moment of the nuclei of leucocytes of Sunset Yellow-treated animals (Fig. 7). Instead, application of Allura Red did not induce any significant genotoxic effects on the leucocytes of rats.

Figure 7 The genetic toxicity of Sunset Yellow and Allura Red on white blood cells of rat.

Bar graph showing the tail moment in nuclei of leucocytes of control, Sunset Yellow-, and Allura Red-treated animals after Comet Assay. Data are Mean ± SD (n = 3, t-test, ∗p < 0.05).

Discussion

The present study investigated the effect of the azo dyes, Sunset Yellow and Allura Red, on the renal and hepatic structure and function, antioxidant status, along with the genotoxic effect on white blood cells of rats. The ALT and AST serum levels increased in rats treated with the separate dyes, indicating obstructive damage in the liver tissue. These findings are corroborated by several recent studies that showed a marked increase in the liver transaminases of animals following ingestion of low doses of synthetic colourants (Sarhan, Shati & Elsaid, 2014; Abd Elhalem et al., 2016; Khayyat et al., 2017). This effect is possibly due to increased cell permeability of hepatocytes. As a result, cytoplasmic enzymes, such as transaminases, leak into the circulation and their activities in the serum increase (Hashem et al., 2010).

Significant elevations in the biochemical parameters of creatinine, urea and uric acid were recorded in the serum of animals treated with Sunset Yellow and Allura Red, respectively, relative to the controls. These biochemical parameters reflect the status of the renal function and can be increased in all forms of kidney injuries (Amin, Abdel Hameid & Abd Elsttar, 2010). Similar observations were previously reported in rats treated with Sunset Yellow (Tawfek et al., 2015; Abd Elhalem et al., 2016), tartrazine (Khayyat et al., 2017) and Fast Green (Ashour & Abdelaziz, 2017). The same parameters can be used to classify a food colourant as a harmful agent, depending on its metabolic activation to form free radicals.

Food azo dyes, such as Sunset Yellow and Allura Red, are metabolised by intestinal bacteria, producing free oxygen radicals (Bansal et al., 2005; Shimada et al., 2010). The formed free radicals can induce lipid peroxidation, which causes cell membrane damage, leading to a cascade of pathological events. We investigated the level of lipid peroxidation by measuring the level of MDA, one of the end-products of lipid peroxidation. Interestingly, the levels of MDA significantly increased in rats treated with Sunset Yellow or Allura Red, indicating an increased oxidative stress due to over-production of reactive oxygen species. Our findings concur with Sarhan, Shati & Elsaid (2014) and Khayyat et al. (2017), who reported an elevation of serum MDA after treatment of rats with food chemical colourants. Moreover, Gil (2014) demonstrated that Sunset Yellow induces the oxidative stress response in H295R cells in vitro.

Our results showed an increase in the level of NO, which is considered as a biomarker of oxidative stress. NO is a free radical that might contribute to alterations in energy metabolism. The excessive production of NO may cause oxidative stress, by forming peroxynitrite with the superoxide anion. Consistent with our data, Peresleni et al. (1996) found that oxidative stress to epithelial cells increases NO production, thereby increasing nitrite formation and decreasing cell viability.

The measurement of antioxidant capacity is an appropriate indication for the total antioxidant defences of various kinds of tissues and organs (Ferrara, Gerber & LeCouter, 2003). Our study showed a reduced antioxidant capacity in rats treated with the dyes that can be attributed to the accelerated production of free radicals during Sunset Yellow or Allura Red metabolism. Likewise, administration of low doses of tartrazine has been shown to induce depletion of the TAC (Khayyat et al., 2017), possibly because of an increased free radical generation or impaired antioxidant machinery, leading to increased oxidative stress. According to Francés et al. (2013), oxidative stress may result from a general increase in the level of reactive oxygen species, a drop in the normal antioxidant systems or both.

The present investigation identified many histopathological and ultrastructural alterations in the liver and kidney of rats treated with Sunset Yellow and Allura Red, individually, and suggest a possible cytotoxic activity of both dyes. Other researchers (Tsuda et al., 2001; Gomes et al., 2013; Bawazir, 2016; Alsolami, 2017) demonstrated similar cytotoxic effects of Sunset Yellow and Allura Red, and their mutagenic action in the liver, kidney, intestinal and germinal cells of animals.

Histopathological examinations revealed alterations in the liver include congestion, fibrosis, leucocytes infiltration, an increased number of Kupffer cells, as well as necrotic and hydropic degeneration in hepatic cells. These changes in the kidney were somewhat comparable to those seen in the liver. Inflammation, necrosis and vacuolisation of the tubular epithelium occurred, in addition to the destruction of glomeruli, with thickening of Bowman’s capsule. The hydropic degeneration of hepatocytes and vacuolisation of tubular cells indicate the occurrence of intracellular oedema, as a result of toxicity or immune aggressions (Kosif et al., 2010; Ramos et al., 2015). Meanwhile, the marked increase in infiltration of leucocytes implies a possible inflammatory response in affected tissues, as noted in earlier works (Soltan & Shehata, 2012; El-Desoky et al., 2017) following exposure to synthetic colouring agents in foods. Deformation of the glomeruli structure may lead to proteinuria, as an intact glomerulus membrane is essential for normal glomerular filtration rate (Nakamura et al., 2004; Louei Monfared, 2013).

The electron micrographs revealed many hepatocytes with degenerated mitochondria, densely packed rough endoplasmic reticulum, lipid droplets, numerous glycogen particles and pyknotic nuclei. In the kidney, proximal tubule cells appeared with pleomorphic vacuoles, unusual mitochondrial arrangement, heterogeneous dense bodies and abnormal nuclei. Many distal tubule cells exhibited destruction of their apical parts, electron-dense mitochondria, numerous lysosomes and atrophic or pyknotic nuclei. In agreement with these outcomes, Bansal et al. (2005) and Bawazir (2016) verified that administration of Sunset Yellow and Allura Red caused destructive changes in hepatocytes and alterations and damage in the kidney of treated animals. Other authors established that the use of food colouring agents might cause necrosis and pathological alterations in the liver, kidney, spleen tissues and brain cell layers of experimental animals (Mahmoud, 2006; Sarkar, 2013; Sarkar & Ghosh, 2017). Damaged liver cells with pyknotic nuclei and disrupted proximal and distal tubules were previously reported in the liver and kidney of rats treated with Metanil Yellow (Sarkar & Ghosh, 2012).

Interestingly, the application of Sunset Yellow or Allura Red alters the expression level of the apoptosis regulatory protein Bcl2 in kidney tissues. Previous experimental data investigating the effect of Sunset Yellow and Allura Red on the levels of Bcl2 are not yet available. In addition to its anti-apoptotic function, Bcl2 promotes cell cycling and increases cell resistance to apoptosis (Huang et al., 1997; Akl et al., 2014). The observed downregulation in the expression levels of Bcl2 may result in increased probability of cell death (Oltvai, Milliman & Korsmeyer, 1993). Bcl2 also regulates normal mitochondrial homeostasis, including the pore permeability of the mitochondrial membranes (Vander Heiden & Thompson, 1999). Hence, the ultrastructural alterations evidenced in the mitochondria may be attributed to the decrease in the Bcl2 levels upon Sunset Yellow or Allura Red treatment. On the contrary, the level of COX2 was elevated upon treatment with Allura Red. It is considered as a pro-inflammatory enzyme, secreted by inflamed tissues (Gilroy et al., 1999), which supports our histopathological observations that indicate inflammatory changes in kidney tissues.

The genotoxic analysis of Sunset Yellow and Allura Red in this work confirmed that Sunset Yellow induced a slight genotoxic effect. Allura Red did not show any remarkable DNA damage in white blood cells, as monitored by the comet assay. Sweeney, Chipman & Forsythe (1994) mentioned that direct-acting oxidative genotoxicity might be induced by azo dyes, including Sunset Yellow, in vitro. Also, Poul et al. (2009) investigated the genotoxic effect of Sunset Yellow, using the gut micronucleus assay in mice after two administrations of 20, 200 or 1,000 mg/kg b wt at 24-h intervals. The authors assessed the genotoxic effects at 24 h after administration of the food colouring agent, by recording the frequency of micronucleated cells, and cell toxicity by identification of the apoptotic and mitotic cells. They found a slight increase in the incidence of micronucleated cells in response to Sunset Yellow application.

Importantly, the metabolic reduction of Sunset Yellow produces sulphonated aromatic amines. The genotoxicity of some of these sulphonated aromatic amines has been reviewed by Jung, Steinle & Anliker (1992), who concluded that some sulphonated aromatic amines possess extremely low genotoxic potential. Sasaki et al. (2002) conducted an in vivo comet assay, to investigate the genotoxic effect of Sunset Yellow in various tissues of mice after gavage with a single dose of two g/kg of the food additive. At 3 and 24 h after administration, Sunset Yellow did not induce genetic aberrations in cells of mice relative to controls. No mutagenic effects were noted in a bone marrow micronucleus assay in vivo after a single oral dose of two g/kg b wt Sunset Yellow (Westmoreland & Gatehouse, 1991).

The in vivo genotoxicity of Allura Red was recently evaluated by Honma (2015) by the induction of DNA damage in the liver and stomach of animals, which concluded that administration of Allura Red was not genotoxic. Another recent study also showed the absence of Allura Red genotoxicity, based on the comet assay, and the in vivo bone marrow micronucleus assay in the liver and colon (Bastaki et al., 2017).

Conclusion

In conclusion, our data show that Sunset Yellow (2.5 mg/kg b wt.) and Allura Red (7 mg/kg b wt.) possess pathological and physiological liver and kidney toxicities in male Wistar albino rats. Sunset Yellow but not Allura Red seems to be slightly genotoxic.

Supplemental Information

Data S1 Raw data of biochemical analysis

Click here for additional data file.

Data S2 Raw data of Comet assay

Click here for additional data file.

Additional Information and Declarations

Competing Interests

Author Contributions

Animal Ethics

Data Availability

The authors declare there are no competing interests.

Latifa I. Khayyat conceived and designed the experiments, authored or reviewed drafts of the paper.

Amina E. Essawy conceived and designed the experiments, performed the experiments, analyzed the data, contributed reagents/materials/analysis tools, prepared figures and/or tables, authored or reviewed drafts of the paper, approved the final draft.

Jehan M. Sorour analyzed the data, contributed reagents/materials/analysis tools, prepared figures and/or tables.

Ahmed Soffar performed the experiments, analyzed the data, contributed reagents/materials/analysis tools, prepared figures and/or tables, authored or reviewed drafts of the paper, approved the final draft.

The following information was supplied relating to ethical approvals (i.e., approving body and any reference numbers):

The experimental procedures were approved by the Menoufia University IACUC committee (Approval No: MNSP155).

The following information was supplied regarding data availability:

The raw data are provided in the Supplemental Files.

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
