# Peer review of "Sunset Yellow and Allura Red modulate Bcl2 and COX2 expression levels and confer oxidative stress-mediated renal and hepatic toxicity in male rats"

_PeerJ, doi:10.7717/peerj.5689_

## Round 0.1 · original submission · Major Revisions

The referees have raised substantial and serious issues about the work presented as well as the way it was presented. Their concerns would warrant rejection; however, I am willing to give the authors the chance to address the concerns in case the authors may have additional data or justification for the use of controls from a prior publication.

Additionally, Referee#1 concerns about the lack of a clear gap of knowledge as well as the unreliability of a single assay for making such conclusions also need to be addressed.

Unless all these concerns are addressed and discussed, we will not be able to reconsider the manuscript.

Reviewer 1 ·

Basic reporting

In this study by Khayyat, et al., the authors attempt to assess the potential adverse health effects caused by two widely used food additives, sunset yellow and allura red. The authors examined the biochemical, histopathological and ultrastructure changes in liver and kidney of adult male albino rats treated with sunset yellow (2.5mg/kg b.wt), or allura red (7mg/kg b.wt) for 30 days. In addition the authors studied the genotoxic effect of both dyes on white blood cells using neutral comet assay. The main findings of this study were that these food additives increased biochemical markers of hepatic and renal function (ALT, AST, urea, uric acid, and creatinine). This study also revealed a noticeable increase in MDA and a marked decrease in total antioxidant levels in animals treated with these food additives as compared to controls. In addition, both dyes had adversely affected the liver and kidney of the treated animals at the histological and fine structure levels. At the molecular level, treatment with these food additives resulted in downregulation of Bcl2 and upregulation of COX2 expression. Comet assay in white blood cells showed no significant DNA damage in leukocytes.

1. The current study presents no novelty to the existing knowledge in the field. It is just an ordinary work which will make no real addition to already existing controversy.

Experimental design

1-The authors did not explicitly mention the gap of knowledge or the scientific void that this study will address.
2-It is not clear why the authors opted to investigate the potential hepatic and renal toxicities of these coloring agents?

Validity of the findings

1-The results of this work are significantly compromised by the notion that authors used the in vivo comet assay as a sole readout of genotoxicity. Indeed, comet assay alone is not enough to ensure reliable detection of relevant in vivo genotoxicants and should be used in conjunction with the micronucleus test in bone marrow and the AMES assay. Furthermore, it is internationally recommended to use the in vivo mammalian alkaline comet assay rather than the neutral comet assay
2- In the materials and methods sections, the authors indicated that Wistar albino adult male rats were treated with either sunset yellow (2.5mg/kg b.wt) or allura red (7mg/kg b.wt). The authors selected the doses based on a previous study by Doguc et al., (2013). Definitely, the referenced study used the indicated doses of sunset yellow or or allura red in different species of rats (Wistar Han strain) with different gender (female rats) and different physiological condition (Pregnancy)! The authors should mention whether theses doses are relevant to the estimated daily intake (EDI) of these agents.

Additional comments

1-The authors stated “our experimental procedures are approved with Menoufia University IACUC” committee’s guide for care of laboratory animal (Approval No: MNSP155). However, the study was not performed in Menoufia University and none of the listed authors is affiliated with the institute?
2- It is occasionally noticeable that the authors are trying to inflate their citation counts through self-citation. This is of particular concern in the materials and methods section, whereas the original articles describing the methods should be cited rather than articles which just used the methods. This improper citation is not a benign practice; adequate and accurate citation is a necessity for scientifically and methodologically sound research.

Reviewer 2 ·

Basic reporting

See combined comments below.

Experimental design

See combined comments below.

Validity of the findings

See combined comments below.

Additional comments

See combined comments below.

Annotated reviews are not available for download in order to protect the identity of reviewers who chose to remain anonymous.

·

Basic reporting

- The manuscript is written in clear professional English.
- provided with enough background information based on literature review related to the subject.
- The manuscript is in acceptable professional structure.
- It is self-contained with relevant results to the hypothesis.
with no comment related to Basic reporting

Experimental design

- It is stated that "the control group has been previously published in our previous work (Khayyat et el., 2017)"., Line 585;as well as in several other places with table1, 6; and figures. Does that mean that the control group is part of the present study design or was actually of a previous work and just included with the present results for comparison. If the later, is the case, that is not acceptable for the proper design of the experiment. If it is part of the present design, it is also unacceptable too, since data are published again even if it is of the control data.
- line 98; is not defined where the experiments are carried out?. "our animal facility"-
- line 102; approval of experimental procedure were taken from Menoufia University, while affiliations of authors are in Alexandria and KSA ???.
Comments on these are strongly recommended.

Validity of the findings

- line 155; Student's t-test is not the proper method to use to analyze data statistically. one way ANOVA with a post-hoc test is advisable to compare the 3 groups.
- line 44; 166; 237; uric acid levels were not statistically significant.
- line 353-355; The conclusion should be more defined to include doses used and duration and be limited to the rat species.
- Legends of Table one should include number of rats used and method of statistical analyses.
Comments on these are strongly recommended.

Additional comments

The manuscript needs major revisions related to experimental design and validity of the findings as previously stated beside few editing, such as: line-42; space before mg; line-48; altered; line-91, we investigated the possible effect of both..; line-166, induced a slight non-significant elevation..;

---

## Round 0.2 · Minor Revisions

The referees have no further scientific issues with the manuscript.

Overall, the manuscript is readable and easy to follow. However, there are still some language issues. The Discussion section seems to be particularly the least revised. Some minor but critical corrections are needed.

I will give a few examples below, and please note that it is not a full list of corrections. ALL OF THE MANUSCRIPT needs to be revised carefully.

* Punctuation needs a quick revision.
Example: Line 156: "however" is a conjunction that starts an independent clause. Thus, the right punctuation is: "was observed; however, a significant elevation ...."
* Line 207: "concerned" the verb is improperly used. The correct use here is: "The present study is concerned"
* Line 212: "studies who recorded": Who is used for humans not studies. Thus it should read: "with previous studies, which"
* Line 223: "as a harmful agent depends on it" ... Bad grammar here.
"depends" is a verb that suggest a new sentence here. The sentences should rather be linked by a gerund, i.e."
"as a harmful agent, depending on its" OR "as a harmful agent in a way that depends ..."
* Line 227: "In consistence" ... What do you mean by this? In consistence with what?
* Line 294: "Our results agrees" one of the two "s"s have to go away! I think you mean "Our results agree". Same comment about Line 219 ("parameters reflects")
* Line 295: "1994, who show" I think you agree with me that 1994 is way into the past. The verb should be "who showed"


AGAIN, the past examples are not exhaustive. Please revise all text.
* * *
* Last but not least, all references should be revised with PeerJ style. I have seen many instances where the year is added between two commas. I'm not sure if this agrees with PeerJ style. Please check.

·

Basic reporting

no comments

Experimental design

their response to the comments are somehow accepted

Validity of the findings

accepted response

Additional comments

comments are corrected

---

## Round 0.3 · Minor Revisions

Unfortunately the manuscript still needs language revision. I have gone quickly through it and consulted the journal office, who also carefully checked the language, and we agreed that the manuscript currently does not meet the language standards of PeerJ.

Different problems with grammar, punctuation, and style make the manuscript hard to follow and affect its clarity. Thus, we recommend that you either have the manuscript revised/edited by some person with expertise in English grammar and technical writing; otherwise you can use the service of any professional editorial service.

We are sorry this will delay the manuscript publication; however, language is a key element in scientific communication and needs to help conveying the message rather than hindering it.

---

## Round 0.4 · accepted · Accept

Thank you very much for taking the editorial comments into consideration, for the comprehensive language revision, and for your patience throughout the process. I believe it paid off at the end as the manuscript looks now way better than the first and second submission.

Best of luck.

#